# Association of Dietary Nutrient Intake with Early Age-Related Macular Degeneration in Japanese-Americans

**DOI:** 10.3390/metabo11100673

**Published:** 2021-09-30

**Authors:** Ayaka Edo, Yunialthy Dwia Pertiwi, Kazuyuki Hirooka, Shun Masuda, Muhammad Irfan Kamaruddin, Masahide Yanagi, Akiko Nagao, Haruya Ohno, Masayasu Yoneda, Yoshiaki Kiuchi

**Affiliations:** 1Department of Ophthalmology and Visual Science, Graduate School of Biomedical and Health Sciences, Hiroshima University, 1-2-3 Kasumi, Minami-Ku, Hiroshima 734-8551, Japan; kazuyk@hiroshima-u.ac.jp (K.H.); masudash@hiroshima-u.ac.jp (S.M.); ykiuchi@hiroshima-u.ac.jp (Y.K.); 2Department of Microbiology, Faculty of Medicine, Hasanuddin University, Jl. Perintis Kemerdekaan No.KM.10, Tamalanrea Indah, Kec. Tamalanrea, Kota Makassar 90245, Sulawesi Selatan, Indonesia; dwiapertiwi@gmail.com; 3Department of Ophthalmology, Faculty of Medicine, Hasanuddin University, Hasanuddin University Hospital Building A 4th Floor, Perintis Kemerdekaan Street Km.11, Makassar 90245, Sulawesi Selatan, Indonesia; iphenk_tampo@yahoo.com; 4Hiroshima-Nomura Eye Clinic, 3-1-202, Matsubara-Cho, Minami-Ku, Hiroshima 732-0822, Japan; myaja111@yahoo.co.jp; 5Division of Nutrient Management, Hiroshima University Hospital, 1-2-3 Kasumi, Minami-Ku, Hiroshima 734-8551, Japan; anagao@hiroshima-u.ac.jp; 6Department of Molecular and Internal Medicine, Graduate School of Biomedical and Health Sciences, Hiroshima University, 1-2-3 Kasumi, Minami-Ku, Hiroshima 734-8551, Japan; haruya-ohno@hiroshima-u.ac.jp (H.O.); masayone17@hiroshima-u.ac.jp (M.Y.)

**Keywords:** age-related macular degeneration, saturated fatty acids, animal fat, nutrient intake

## Abstract

Lifestyle factors may be associated with the development of age-related macular degeneration (AMD), in addition to demographic and genetic factors. The purpose of this cross-sectional study is to elucidate the association between nutrient intake and AMD in the Japanese-American population living in Los Angeles. We conducted a medical survey of Japanese immigrants and their descendants living in Los Angeles, including interviews on dietary habits, fundus photography, and physical examinations. Participants were classified into early AMD and control groups on the basis of fundus photographic findings. Consequently, among the 555 participants, 111 (20.0%) were diagnosed with early AMD. There were no late-stage AMD participants. Multivariate logistic regression analysis showed that the intake of animal fat and saturated fatty acids (SFA) was positively associated with early AMD (*p* for trend = 0.01 for animal fat, *p* for trend = 0.02 for SFA), and the intake of vegetable fat, total carbohydrate, simple carbohydrate, sugar, and fructose was inversely associated with early AMD (*p* for trend = 0.04 for vegetable fat, *p* for trend = 0.046 for carbohydrate, *p* for trend = 0.03 for simple carbohydrate, *p* for trend = 0.046 for sugar, *p* for trend = 0.02). Our findings suggest that excessive animal fat and SFA intake increases the risk for early AMD in Japanese-Americans whose lifestyles have been westernized.

## 1. Introduction

Age-related macular degeneration (AMD) is one of the major causes of visual impairment and irreversible blindness among the elderly, and its prevalence is projected to increase in the future [1]. Its development is due to a complex interaction between demographic, genetic, and environmental factors. While age, race, and genetic factors strongly influence AMD development and progression, modifiable lifestyle factors, including smoking history and nutrient intake, have also been shown to be associated with AMD [2].

It has been suggested that dietary factors may influence AMD development and progression [2,3]. Nutrients that affect AMD susceptibility include carotenoids (lutein, zeaxanthin, β-carotene), vitamins (vitamins A, C, E), mineral supplements (zinc), and dietary fats (saturated fatty acids [SFA], polyunsaturated fatty acids [PUFA], cholesterol) [3]. Further research is needed to elucidate the role of individual nutrients and supplements on a patient’s risk for AMD development and progression.

Some racial differences in AMD patterns and genotypes associated with AMD have been reported [4,5]. Regarding AMD patterns, Japanese individuals are reported to have a higher incidence of polypoidal choroidal vasculopathy type than Caucasians [4]. Ethnic genotypic variation has been reported with AMD-associated complement factor H (*CFH*) polymorphisms. Although a significant association between *CFH* Y402H and AMD has been reported in Westerners and Chinese individuals [6,7,8], several studies have failed to demonstrate a significant association between Y402H and AMD in Japanese individuals [9,10,11]. In addition, previous reports have suggested that the effects of nutritional intake on AMD vary with genetic background [12,13]. For instance, in one study higher intake of lutein/zeaxanthin reduced the risk of early AMD in participants at high genetic risk of AMD, whereas weekly consumption of fish reduced the risk of late AMD. However, no similar associations have been found among individuals at low genetic risk [13]. Although Japanese-Americans have Japanese genetic predispositions, they live in an American environment with corresponding dietary and exercise habits [14]. Therefore, it is possible that the association between nutritional intake and AMD in Japanese-Americans may differ from that in native Japanese or Americans.

Every few years since 1970, we have conducted the medical examination of Japanese-Americans in an epidemiological study called the Hawaii–Los Angeles–Hiroshima study [14,15]. In 2015, we performed dietary assessment and took fundus photographs of Japanese-American participants living in Los Angeles, California. The purpose of this study is to investigate the association between nutrient intake and early AMD in Japanese-Americans living in Los Angeles, California.

## 2. Results

Of 584 participants, 18 were excluded owing to ungradable retinal image quality because of poor camera focus or ocular media opacity. Subsequently, 11 participants with incomplete smoking history, physical, or dietary data were excluded, leaving 555 patients who were included in the analysis (Figure 1).

The baseline characteristics of the study participants are presented in Table 1. Early AMD, which is characterized mainly by large soft drusen or RPE abnormalities, was diagnosed in 111 participants (20.0%). There were no participants with late AMD. The mean ± standard deviation (SD) for age was 66.5 ± 10.3 years in the early AMD group and 60.4 ± 13.7 years in the control group, which was significantly higher in the early AMD group (*p* < 0.001). No statistically significant differences were found for sex (*p* = 0.60), body mass index (BMI) (*p* = 0.73), smoking history (*p* = 0.46), hypertension (*p* = 0.16), diabetes (*p* = 0.06), or generation (*p* = 0.75) between the two groups. Table 2 shows the daily dietary nutrient intake of the early AMD and control groups. The mean animal fat intake was significantly higher in the AMD group compared to that of the control group (*p* = 0.03). By contrast, the mean vegetable fat and PUFA intake was significantly lower in the AMD group than in the control group (*p* = 0.03 for vegetable fat; *p* = 0.007 for PUFA).

Table 3 and Table 4 show the results of the multivariate logistic regression analysis assessing the association between dietary macro- and micro-nutrient intake and early AMD after adjusting for age, sex, smoking history, hypertension, and diabetes. The animal fat and SFA intake were found to be positively associated with early AMD (odds ratios [OR] for the second, third, and highest quartiles: 1.12 [95% confidence interval [CI], 0.59–2.13], 0.88 [0.45–1.72], 1.86 [1.01–3.42], *p* value for trend = 0.01 for animal fat;1.27 [0.66–2.44], 1.14 [0.58–2.23], 1.80 [0.96–3.40], *p* value for trend = 0.02 for SFA). On the other hand, the vegetable fat, total carbohydrate, simple carbohydrate, sugar, and fructose intake were negatively associated with early AMD (odds ratios [OR] for the second, third, and highest quartiles: 0.71 [95% CI, 0.39–1.26], 0.53 [0.29–0.99], 0.54 [0.29–1.01], *p* value for trend = 0.04 for vegetable fat; 0.68 [0.38–1.23], 0.61 [0.34–1.11], 0.58 [0.31–1.05], *p* value for trend = 0.046 for total carbohydrate; 0.51 [0.28–0.92], 0.60 [0.34–1.07], 0.54 [0.30–0.97], *p* value for trend = 0.03 for simple carbohydrate; 0.56 [0.31–1.01], 0.66 [0.37–1.17], 0.54 [0.30–0.99], *p* value for trend = 0.046 for sugar; 1.27 [0.70–2.32], 0.83 [0.45 –1.54], 0.59 [0.31–1.13], *p* value for trend =0.02 for fructose). There was no significant association of animal protein, vegetable protein, PUFA, cholesterol, complex carbohydrate, fiber, vitamin A, vitamin B_1_, vitamin B_2_, vitamin C, calcium, iron, potassium, or salt with early AMD.

## 3. Discussion

In the current study, we demonstrated positive associations between animal fat and SFA intake and early AMD, whereas we found negative associations between vegetable fat, total carbohydrate, simple carbohydrate, sugar, and fructose intake and early AMD in Japanese-Americans. 

Previous reports on westerners demonstrated a positive association between SFA intake and AMD [16,17,18]. However, the Tsuruoka Metabolomics Cohort Study of native Japanese subjects reported results that were contrary to ours, whereby the SFA intake reduced the prevalence of early AMD [19]. The participants in this study consisted of immigrants from Japan and their descendants with Japanese genetic predispositions [14]. We consider that the difference in the baseline SFA intake may be the reason for the different results from those of the Tsuruoka Metabolomics Cohort Study. It is known that the SFA intake is higher in westerners than in Asians because of differences in dietary styles [20]. Japanese-Americans are highly exposed to westernized lifestyles and ingest more fat than native Japanese [21]. Indeed, in the present study, the median quartiles of energy-adjusted SFA intake were 13.8, 18.8, 22.5, and 28.7 g/day, in the order from the lowest to the highest, respectively. By contrast, in the Tsuruoka Metabolomics Cohort Study, the median quartiles of energy-adjusted SFA intake were 8.7, 10.4, 12.1, and 15.1 g/day, in the order from the lowest to the highest, respectively. Our study subjects ingested approximately twice as much SFA as those in the Tsuruoka Metabolomics Cohort Study [19]. Taking these differences into consideration, we assume that there is probably an optimal SFA intake, and both, too much and too little intake may have negative effects on the retina. Retinal pigment epithelium (RPE) cells are responsible for lipid metabolism in the retina. While lipids are important as an energy source for the retina, excess amounts are secreted by RPE cells [22]. The deposited lipids induce oxidative stress and inflammation, resulting in RPE dysfunction [5]. Indeed, the presence of drusen, which are subretinal deposits of lipids, are the prominent characteristic lesions of AMD.

Our epidemiological results on the association between SFA intake and AMD are supported by the results of experimental studies. One of the pathogenic mechanisms of AMD is the dysfunction and degeneration of the RPE due to oxidative stress and chronic inflammation [23]. Montserrat-De La Paz et al. found that oxidative stress and chronic inflammation were induced in RPE cells exposed to dietary fat, particularly SFA [24]. In animal experiments, Nagai et al. demonstrated that high-fat diet-fed mice accumulate abnormal lipid deposition in the RPE, resulting in visual impairment, and have been used as an AMD model [25].

In the present study, there was no significant association between dietary PUFA intake and early AMD. PUFA can be broadly classified into ω-3-PUFA and ω-6 PUFA [26]. ω-3 PUFA are thought to have a protective effect on the retina because they contain docosahexaenoic acid and eicosapentaenoic acid, which are important for photoreceptor survival and have anti-inflammatory effects [27,28]. Several epidemiological studies reported that a high ω-3 PUFA intake significantly reduced the risk for AMD [29,30,31,32], and the same result was shown in Japanese [33]. By contrast, ω-6 PUFA have a pro-inflammatory effect [34]. Seddon et al. reported that a high ω-6 PUFA intake significantly increased the risk for AMD [35]. In our study, we were unable to obtain individual intake data for each PUFA type, which may have prevented us from fully investigating the association between PUFA and AMD.

Our findings indicate that animal fat intake is positively associated, whereas vegetable fat intake is negatively associated, with early AMD. In the USA, the main source of animal fat is red meat, and animal fat is rich in SFA [36,37]. High consumption of olive oil and nuts reportedly has a protective effect against AMD [38,39]. Considering these reports, our results seem to be reasonable. However, few published studies have examined associations between animal or vegetable fat intake and AMD risk. Sedden et al. showed a significant association between high vegetable fat intake and advanced AMD [35]. Another cohort study of patients with intermediate AMD, higher intake of vegetable fat, and to a lesser extent animal fat, had an increased risk for neovascular AMD [38]. Regarding the association between vegetable fat and AMD, our results are not consistent with the above findings. The stage of AMD and age and race of the participants differed between our study and the above-cited studies. Furthermore, dietary fat is typically a mixture of different types of fatty acids. Some animal fat contains more monounsaturated fatty acids than SFA, and concentrations of both ω-3 and ω-6 PUFA vary between different vegetable fats [36]. Variations in the proportions of the individual fatty acids that make up animal and vegetable fats related to differing lifestyle of participants may account for these inconsistencies.

In our study, we demonstrated negative associations between total carbohydrate, simple carbohydrate, sugar, and fructose intake and early AMD. Several studies have investigated the role of carbohydrate in AMD risk by using the glycemic index (GI) that is the measure that indicates how fast blood glucose is raised after consuming a carbohydrate-containing food. These studies showed an effective role of a low dietary glycemic index in preventing the progression of AMD [3,40,41,42]. On the other hand, multivariate analysis of the association between carbohydrate intake and neovascular AMD showed that, although not significant, the odds ratios for neovascular AMD were lower in quintile 2, 3, 4, and 5 than in quintile 1 with the lowest carbohydrate intake, respectively [41]. High GI diets cause a rapid increase in blood glucose during the postprandial period. This results in higher concentrations of glucose entering the cells, resulting in chronically high oxidative stress. It has been suggested that low GI diets may more effectively reduce such oxidative stress than low carbohydrate diets [43]. Sugar is classified as medium GI and fructose as low GI [44,45]. Thus, considering that neither sugar nor fructose is classified as high GI, our results for simple carbohydrate, sugar, and fructose may be consistent with previously reported findings. However, we cannot simply conclude from our findings that higher total carbohydrate, simple carbohydrate, sugar, and fructose intake more effectively prevents early AMD. Further studies, including GI and sources of relevant nutrients, are needed to clarify the relationships between total carbohydrate, simple carbohydrate, sugar, and fructose intake and early AMD.

Because oxidative damage in the retina is a key process involved in AMD development, it is considered that dietary and supplementary antioxidants prevent AMD [3]. The Age-Related Eye Disease Study (AREDS) showed that the vitamin A intake was significantly associated with a lower risk for late AMD [18]. Conversely, some studies showed that there was no significant inverse association between vitamin A and AMD [46,47]. Vitamin C is a well-known antioxidant that prevents free radical and reactive oxygen species damage [3]. The beneficial effect of vitamin C to prevent early AMD is supported by Gopineth et al. and Aoki et al. [33,48]. However, the meta-analysis by Chong et al. concluded that vitamin C has little or no such effect [49]. Our study also did not find a significant inverse association between vitamin A or C intake and early AMD. The AREDS suggests that a high intake of vitamins C, E, and β-carotene alone is insufficient to prevent AMD and that the addition of zinc is necessary [50]. These inconsistent results need to be repeated on a larger scale, and including other antioxidant intake, such as carotenoids, vitamin E, zinc, and copper.

Our study has several limitations. First, the association between dietary nutrition and AMD was only analyzed from temporal information because this was a cross-sectional study. Although AMD is a multifactorial disease that might be affected by long-term dietary status, the current study was only able to assess nutritional status at one time point. Second, the number of participants was limited and the average age of the participants was in the 60s, which is relatively young to evaluate AMD. This resulted in a small number of case groups. Third, the nutrients analyzed were limited. We could not examine the origin of the nutrients, including those such as ω-3 PUFA, carotenoids, zinc, or copper, which previous reports have shown to be associated with a reduced risk for AMD [3,18,29,30,31,32,48,50,51]. Forth, participants’ background factors and dietary intake information were self-reported. Errors in the measurement of dietary intake and the covariates might have limited our ability to obtain accurate odds ratios estimates. Fifth, subtle signs of AMD that were not detected by fundus photography may have been missed and the patients may have been assigned to the control group because fundus photography was performed as a non-stereoscopic evaluation. Future studies will need to address these issues and conduct prospective studies with larger sample sizes to determine the causal relationship between nutrient intake and AMD.

In conclusion, we have shown that the high intake of animal fat and SFA is significantly associated with early AMD, and the vegetable fat intake was inversely associated with early AMD in Japanese-Americans who currently live in Los Angeles. This suggests that an excessive SFA intake may promote AMD development.

## 4. Materials and Methods

### 4.1. Study Participants

The present cross-sectional study was conducted with 584 Japanese-American subjects who participated in the medical survey in Los Angeles in August 2015 as part of the Hawaii–Los Angeles–Hiroshima study [14,15]. All participants were Japanese-Americans living in the suburbs of Los Angeles and were immigrants from Japan and their descendants. The participants underwent physical, dietary, and fundus photographic examinations by well-trained dietitians, internists, optometrists, and nurses.

All subjects provided written informed consent to participate in the examinations. This study was approved by the Hiroshima University Ethics Committee (approval No. E-139) in accordance with the provisions of the Declaration of Helsinki.

### 4.2. Dietary Assessment

Dietary information was collected using the food frequency method as previously described [14,52]. Briefly, the frequency of intake, the amount of intake per meal, and the cooking method of each food group were ascertained using a food model by a dietitian in a personal interview, and the values for daily total energy and intake for individual nutritional elements (i.e., animal protein, vegetable protein, animal fat, vegetable fat, SFA, PUFA, cholesterol, total carbohydrate, simple carbohydrate, complex carbohydrate, sugar, fructose, fiber, vitamins A, B_1_, B_2_, and C, calcium, iron, potassium, and salt) were calculated. The mean daily intake of each food group was calculated as (mean intake per meal) × (frequency of intake per day), and the nutritional intake from each food group was calculated as (nutritional value per gram of each food) × (mean daily intake of each food group) [14,52]. The nutritional value of each food group was determined on the basis of the U.S. Department of Agriculture’s Nutritive Value of American Foods in Common Units [53]. Intake of simple carbohydrate was defined as the sum of intake of sugar and fructose. Nutrient intake used in the analysis was adjusted for total energy intake using the residual method as previously described [54].

### 4.3. Evaluation of Fundus Photographs

The participants underwent bilateral fundus photography under non-mydriatic conditions using a 45 degree nonmydriatic retinal camera (NIDEK AFC-300; NIDEK Co., Ltd., Gamagori, Japan). The fundus photographs taken were centered on the macula. The fundus images were evaluated by ophthalmologists masked to the subject information, using a non-stereoscopic retinal image evaluation protocol in accordance with the Wisconsin Age-Related Maculopathy Grading System as described previously [55,56,57]. Briefly, the fundus photograph of the right eye was used, unless the image could not be evaluated, in which case, the photograph of the left eye was used.

According to the disease severity, AMD was categorized as early or late AMD. Early AMD was defined as the absence of signs of late AMD, and the presence of large soft drusen (distinct drusen or indistinct drusen) >125 μm in diameter and/or RPE abnormalities (hypo- or hyperpigmentation) within a grading grid (within a radius of 3000 μm centered on the fovea) [19,58,59]. Late AMD was defined as the presence of neovascular AMD or geographic atrophy. Neovascular AMD was defined as lesions exhibiting subretinal or sub-RPE hemorrhage, RPE detachment, serous detachment of the sensory retina, or subretinal fibrous scarring [58]. Geographic atrophy was characterized by atrophic areas with a sharp border and visible choroidal vessels [58].

### 4.4. Assessment of Other Variables

Each participant underwent a comprehensive evaluation, including a physical examination and an interview about their life and medical history after an overnight fast with written informed consent. Blood pressure was measured for systolic and diastolic pressure. Hypertension was determined by the mean arterial blood pressure (MABP), which was calculated as one-third systolic blood pressure plus two-thirds diastolic blood pressure. Based on previous reports, a MABP ≥ 105.68 mmHg or being under treatment for hypertension was defined as hypertension [60]. The BMI was calculated as weight (kg)/height squared (m^2^). The participants without diabetes underwent fasting serum glucose measurement and a 75-g oral glucose tolerance test (OGTT). Diabetes was defined as either previous hospital diagnosis, a fasting serum glucose level ≥126 mg/dL, or a 2-hour serum glucose after OGTT level ≥200 mg/dL, based on the American Diabetes Association guidelines [61]. On the basis of the participants’ self-reports, smoking history was classified into three categories: nonsmoker, former smoker, and current smoker.

### 4.5. Statistical Analysis

Because there were no late AMD subjects, the participants were classified into the early AMD group and the control group according to the findings of fundus photographs. Demographic factors in the early AMD and control groups were compared using the Wilcoxon rank sum test for continuous variables and the chi-square test for categorical variables. Continuous variables are presented as mean ± SD. Subsequently, multivariate logistic regression analysis was performed to examine the nutrient intake associated with early AMD development. Quartiles were used to examine the OR and CI of early AMD risk between different energy-adjusted nutrient intake levels, with the lowest quartile analyzed as the reference group. The following factors, known to be confounders of AMD, were adjusted for the analysis: age (years, continuous), gender (male/female), smoking history (never/former/current), BMI (kg/m^2^, continuous), hypertension (yes/no), and diabetes (yes/no). The first model was adjusted for age and diabetes; the second model was adjusted for age, sex, smoking history, BMI, hypertension, and diabetes. *p* for trend was calculated using nutrient intake as a continuous variable. All statistical analyses were performed using JMP pro 15.0.0 (SAS Institute Inc., Cary, NC, USA). A *p* value < 0.05 was considered statistically significant.

## Figures and Tables

**Figure 1 metabolites-11-00673-f001:**
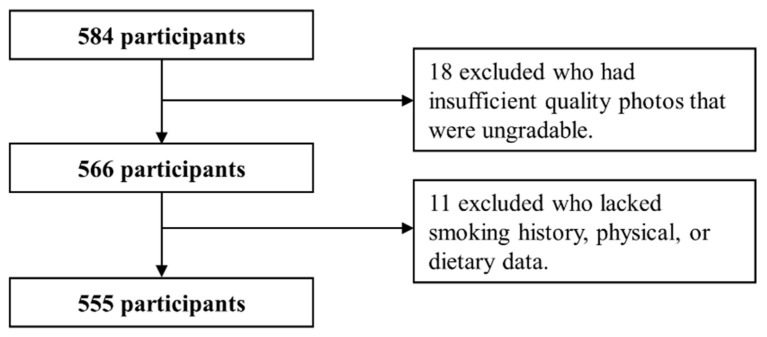
Flow chart demonstrating the identification of the study participants.

**Table 1 metabolites-11-00673-t001:** Clinical characteristics of study participants.

Characteristics	Early AMD (n = 111)	Control (n = 444)	*p* Value ^1^
Age (years)	66.5 ± 10.3	60.4 ± 13.7	<0.001
Sex (M/F)	41/70	176/268	0.60
BMI (kg/m^2^)	23.2 ± 3.7	23.3 ± 3.7	0.73
Smoking, n (%)			0.46
Never	71 (64.0%)	256 (57.7%)	
Former	30 (27.0%)	136 (30.6%)	
Current	10 (9.0%)	52 (11.7%)	
Hypertension, n (%)	24 (21.6%)	71 (16.0%)	0.16
Diabetes, n (%)	15 (13.5%)	35 (7.9%)	0.06
Generation, n (%)			0.75
1st	97 (87.4%)	381 (85.8%)	
2nd	5 (4.5%)	30 (6.8%)	
3rd	8 (7.2%)	24 (5.4%)	
4th	1 (0.9%)	8 (1.8%)	
5th	0 (0%)	1 (0.2%)	

^1^ Continuous variables were evaluated using the Wilcoxon rank-sum test while categorical variables were evaluated using the chi-squared test. The difference was considered significant at *p* < 0.05. AMD, age-related macular degeneration; M, male; F, female; BMI, body mass index. All values are presented as mean ± SD or n (%).

**Table 2 metabolites-11-00673-t002:** Daily energy-adjusted nutrient intake of control and early age-related macular degeneration participants.

Nutrient	Early AMD (n = 111)	Control (n = 444)	*p* Value ^1^
Total energy (kcal)	2249.5 ± 684.8	2220.2 ± 607.1	0.86
Animal protein	36.7 ± 11.3	37.8 ± 10.9	0.31
Vegetable protein	34.6 ± 7.8	35.1 ± 7.0	0.57
Animal fat (g)	44.5 ± 36.2	35.9 ± 26.2	0.03
Vegetable fat (g)	34.0 ± 10.5	35.9 ± 8.1	0.03
SFA (g)	23.3 ± 10.3	21.0 ± 7.7	0.06
PUFA (g)	13.7 ± 3.5	14.4 ± 2.8	0.007
Cholesterol (mg)	274.8 ± 91.2	279.6 ± 95.9	0.91
Total carbohydrate (g)	280.1 ± 70.6	293.1 ± 58.4	0.08
Simple carbohydrate (g)	255.0 ± 62.0	267.1 ± 51.0	0.07
Complex carbohydrate (g)	25.1 ± 11.1	26.0 ± 10.6	0.75
Sugar (g)	252.2 ± 62.7	263.4 ± 51.5	0.09
Fructose (g)	2.8 ± 3.6	3.7 ± 4.2	0.23
Fiber (g)	4.3 ± 1.8	4.3 ± 1.8	0.62
Vitamin A (μgRAE)	597.6 ± 255.6	599.3 ± 257.7	0.79
Vitamin B_1_ (mg)	0.91 ± 0.27	0.92 ± 0.28	0.89
Vitamin B_2_ (mg)	1.23 ± 0.37	1.23 ± 0.34	0.74
Vitamin C (mg)	167.8 ± 71.7	169.6 ± 69.0	0.90
Calcium (mg)	636.2 ± 201.9	602.7 ± 176.3	0.10
Iron (mg)	6.9 ± 1.9	7.0 ± 2.1	0.92
Potassium (mg)	2924.5 ± 748.8	2949.3 ± 729.9	0.89
Salt (g)	5.4 ± 2.5	5.7 ± 2.3	0.30

^1^ Wilcoxon rank-sum test; the difference was considered significant at *p* < 0.05. AMD, age-related macular degeneration; SFA, saturated fatty acids; PUFA, polyunsaturated fatty acids; RAE, retinol activity equivalents. All values are presented as mean ± SD. Nutrient intake used energy-adjusted values.

**Table 3 metabolites-11-00673-t003:** Associations between energy-adjusted macronutrient intake and early AMD in multivariate logistic regression analysis.

Nutrient	Odds Ratio (95% CI)				*p* for Trend
	Q1 (Lowest)	Q2	Q3	Q4 (Highest)	
**Animal protein (g)**	≤30.3	30.3–36.8	36.8–42.9	>42.9	
No. with outcome/at risk	34/139 (24.5%)	28/139 (20.1%)	18/139 (13.0%)	31/138 (22.5%)	
Model 1 ^1^	1 (Reference)	0.70 (0.39–1.26)	0.47 (0.25–0.89)	0.91 (0.51–1.61)	0.42
Model 2 ^2^	1 (Reference)	0.69 (0.39–1.24)	0.47 (0.25–0.89)	0.92 (0.52–1.65)	0.22
**Vegetable protein (g)**	≤30.6	30.6–34.9	34.9–39.6	>39.6	
No. with outcome/at risk	34/139 (24.5%)	24/139 (17.3%)	24/139 (17.3%)	29/138 (21.0%)	
Model 1 ^1^	1 (Reference)	0.55 (0.30–1.00)	0.55 (0.30–1.00)	0.69 (0.39–1.24)	0.25
Model 2 ^2^	1 (Reference)	0.54 (0.29–0.99)	0.54 (0.29–0.99)	0.68 (0.38–1.23)	0.27
**Animal fat (g)**	≤22.3	22.3–33.2	33.2–46.2	>46.2	
No. with outcome/at risk	22/139 (15.8%)	27/139 (19.4%)	23/139 (16.6%)	39/138 (28.3%)	
Model 1 ^1^	1 (Reference)	1.12 (0.59–2.11)	0.93 (0.48–1.78)	1.88 (1.03–3.42)	0.01
Model 2 ^2^	1 (Reference)	1.12 (0.59–2.13)	0.88 (0.45–1.72)	1.86 (1.01–3.42)	0.01
**Vegetable fat (g)**	≤29.7	29.7–35.5	35.5–40.6	>40.6	
No. with outcome/at risk	35/139 (25.2%)	30/139 (21.6%)	24/139 (17.3%)	22/138 (15.9%)	
Model 1 ^1^	1 (Reference)	0.74 (0.42–1.31)	0.56 (0.31–1.03)	0.55 (0.30–1.01)	0.03
Model 2 ^2^	1 (Reference)	0.71 (0.39–1.26)	0.53 (0.29–0.99)	0.54 (0.29–1.01)	0.04
**SFA (g)**	≤17.0	17.0–20.7	20.7–24.5	>24.5	
No. with outcome/at risk	20/139 (14.4%)	29/139 (20.9%)	27/139 (19.4%)	35/138 (25.4%)	
Model 1 ^1^	1 (Reference)	1.31 (0.69–2.49)	1.19 (0.62–2.29)	1.85 (0.99–3.44)	0.02
Model 2 ^2^	1 (Reference)	1.27 (0.66–2.44)	1.14 (0.58–2.23)	1.80 (0.96–3.40)	0.02
**PUFA (g)**	≤12.4	12.4–13.9	13.9–15.9	>15.9	
No. with outcome/at risk	36/139 (25.9%)	32/140 (22.9%)	21/138 (15.2%)	22/138 (15.9%)	
Model 1 ^1^	1 (Reference)	0.81 (0.46–1.42)	0.49 (0.27–0.91)	0.56 (0.31–1.04)	0.07
Model 2 ^2^	1 (Reference)	0.78 (0.44–1.37)	0.48 (0.26–0.89)	0.56 (0.30–1.04)	0.08
**Cholesterol (mg)**	≤218.7	218.7–267.0	267.0–322.3	>322.3	
No. with outcome/at risk	23/139 (16.5%)	31/139 (22.3%)	31/139 (22.3%)	26/138 (18.8%)	
Model 1 ^1^	1 (Reference)	1.59 (0.86–2.93)	1.52 (0.82–2.81)	1.17 (0.62–2.20)	0.55
Model 2 ^2^	1 (Reference)	1.52 (0.82–2.83)	1.52 (0.82–2.83)	1.14 (0.61–2.16)	0.55
**Total carbohydrate (g)**	≤266.4	266.4–295.3	295.3–323.4	>323.4	
No. with outcome/at risk	36/139 (25.9%)	27/139 (19.4%)	25/139 (18.0%)	23/138 (16.7%)	
Model 1 ^1^	1 (Reference)	0.68 (0.38–1.21)	0.61 (0.34–1.10)	0.56 (0.21–1.03)	0.04
Model 2 ^2^	1 (Reference)	0.68 (0.38–1.23)	0.61 (0.34–1.11)	0.58 (0.31–1.05)	0.046
**Simple carbohydrate (g)**	≤244.0	244.0–268.8	268.8–293.7	>293.7	
No. with outcome/at risk	38/139 (27.3%)	23/139 (16.6%)	26/139 (18.7%)	24/138 (17.4%)	
Model 1 ^1^	1 (Reference)	0.51 (0.28–0.92)	0.59 (0.33–1.06)	0.53 (0.29–0.96)	0.03
Model 2 ^2^	1 (Reference)	0.50 (0.27–0.91)	0.60 (0.34–1.07)	0.54 (0.30–0.97)	0.03
**Complex carbohydrate (g)**	≤18.8	18.8–25.7	25.7–32.1	>32.1	
No. with outcome/at risk	29/139 (20.9%)	26/139 (18.7%)	30/139 (21.6%)	26/138 (18.8%)	
Model 1 ^1^	1 (Reference)	0.88 (0.48–1.61)	1.10 (0.61–1.97)	0.96 (0.53–1.76)	0.68
Model 2 ^2^	1 (Reference)	0.89 (0.49–1.63)	1.12 (0.62–2.03)	0.99 (0.54–1.82)	0.73
**Sugar (g)**	≤239.3	239.3–265.8	265.8–291.6	>291.6	
No. with outcome/at risk	37/139 (26.6%)	24/140 (17.1%)	27/139 (19.4%)	24/137 (16.8%)	
Model 1 ^1^	1 (Reference)	0.56 (0.31–1.00)	0.65 (0.36–1.16)	0.54 (0.29–0.98)	0.04
Model 2 ^2^	1 (Reference)	0.56 (0.31–1.01)	0.66 (0.37–1.17)	0.54 (0.30–0.99)	0.046
**Fructose (g)**	≤0.1	0.1–1.3	1.3–6.8	>6.8	
No. with outcome/at risk	28/142 (19.7%)	36/136 (26.5%)	26/139 (18.7%)	21/138 (15.2%)	
Model 1 ^1^	1 (Reference)	1.30 (0.73–2.32)	0.85 (0.46–1.56)	0.60 (0.32–1.14)	0.02
Model 2 ^2^	1 (Reference)	1.27 (0.70–2.32)	0.83 (0.45–1.54)	0.59 (0.31–1.13)	0.02
**Fiber (g)**	≤3.2	3.2–4.0	4.0–5.1	>5.1	
No. with outcome/at risk	30/139 (21.6%)	27/139 (19.4%)	30/139 (21.6%)	24/138 (17.4%)	
Model 1 ^1^	1 (Reference)	0.86 (0.47–1.56)	0.92 (0.51–1.66)	0.75 (0.41–1.38)	0.63
Model 2 ^2^	1 (Reference)	0.89 (0.49–1.63)	0.94 (0.52–1.70)	0.73 (0.40–1.36)	0.62

^1^ Adjusted for age and diabetes. ^2^ Adjusted for age, sex, smoking history, body mass index, hypertension, and diabetes. SFA, saturated fatty acids; PUFA, polyunsaturated fatty acids; CI, confidence interval. Nutrient intake was used with energy-adjusted values.

**Table 4 metabolites-11-00673-t004:** Associations between energy-adjusted micronutrient intake and early AMD in multivariate logistic regression analysis.

Nutrient	Odds Ratio (95% CI)				*p* for Trend
	Q1 (Lowest)	Q2	Q3	Q4 (Highest)	
**Vitamin A (μgRAE)**	≤423.4	423.4–525.2	525.2–729.1	>729.1	
No. with outcome/at risk	32/139 (23.0%)	21/139 (15.1%)	27/139 (19.4%)	31/138 (22.4%)	
Model 1 ^1^	1 (Reference)	0.54 (0.29–1.00)	0.73 (0.40–1.31)	0.89 (0.50–1.59)	0.82
Model 2 ^2^	1 (Reference)	0.55 (0.29–1.04)	0.74 (0.41–1.35)	0.88 (0.49–1.58)	0.77
**Vitamin B_1_ (g)**	≤0.73	0.73–0.88	0.88–1.1	>1.1	
No. with outcome/at risk	28/141 (19.9%)	27/140 (19.3%)	25/141 (17.7%)	31/133 (23.3%)	
Model 1 ^1^	1 (Reference)	0.92 (0.51–1.68)	0.90 (0.49–1.67)	1.32 (0.73–2.39)	0.83
Model 2 ^2^	1 (Reference)	0.94 (0.51–1.72)	0.95 (0.51–1.77)	1.39 (0.76–2.53)	0.94
**Vitamin B_2_ (g)**	≤1.0	1.0–1.2	1.2–1.5	>1.5	
No. with outcome/at risk	29/139 (20.9%)	23/143 (16.1%)	24/136 (17.7%)	35/137 (25.6%)	
Model 1 ^1^	1 (Reference)	0.66 (0.36–1.23)	0.71 (0.38–1.32)	1.23 (0.69–2.19)	0.97
Model 2 ^2^	1 (Reference)	0.67 (0.36–1.25)	0.72 (0.39–1.34)	1.27 (0.70–2.28)	0.96
**Vitamin C (mg)**	≤119.3	119.3–155.6	155.6–206.0	>206.0	
No. with outcome/at risk	29/139 (20.9%)	26/139 (18.7%)	32/139 (23.0%)	24/138 (17.4%)	
Model 1 ^1^	1 (Reference)	0.75 (0.41–1.38)	0.98 (0.55–1.76)	0.70 (0.38–1.30)	0.58
Model 2 ^2^	1 (Reference)	0.78 (0.42–1.46)	1.00 (0.55–1.82)	0.69 (0.37–1.29)	0.54
**Calcium (g)**	≤473.7	473.7–605.1	605.1–729.5	>729.5	
No. with outcome/at risk	29/139 (20.9%)	20/139 (14.4%)	23/139 (16.6%)	39/138 (28.3%)	
Model 1 ^1^	1 (Reference)	0.55 (0.29–1.05)	0.64 (0.34–1.20)	1.20 (0.67–2.12)	0.34
Model 2 ^2^	1 (Reference)	0.54 (0.28–1.03)	0.61 (0.32–1.15)	1.21 (0.68–2.17)	0.32
**Iron (g)**	≤5.4	5.4–6.7	6.7–8.2	>8.2	
No. with outcome/at risk	26/140 (18.6%)	25/138 (18.1%)	34/139 (24.5%)	26/138 (18.8%)	
Model 1 ^1^	1 (Reference)	0.95 (0.51–1.77)	1.32 (0.73–2.38)	1.00 (0.54–1.83)	0.66
Model 2 ^2^	1 (Reference)	0.97 (0.52–1.82)	1.36 (0.75–2.47)	1.00 (0.54–1.85)	0.67
**Potassium (mg)**	≤2413.2	2413.2–2822.2	2822.2–3341.8	>3341.8	
No. with outcome/at risk	33/139 (23.7%)	21/139 (15.1%)	28/139 (20.1%)	29/138 (21.0%)	
Model 1 ^1^	1 (Reference)	0.51 (0.27–0.94)	0.72 (0.40–1.28)	0.76 (0.43–1.37)	0.57
Model 2 ^2^	1 (Reference)	0.52 (0.28–0.97)	0.73 (0.40–1.33)	0.75 (0.42–1.35)	0.57
**Salt (g)**	≤3.9	3.9–5.4	5.4–7.2	>7.2	
No. with outcome/at risk	31/140 (22.1%)	27/138 (19.6%)	27/140 (19.3%)	26/137 (19.0%)	
Model 1 ^1^	1 (Reference)	0.97 (0.53–1.75)	0.86 (0.48–1.55)	0.89 (0.49–1.62)	0.36
Model 2 ^2^	1 (Reference)	0.98 (0.54–1.79)	0.86 (0.47–1.56)	0.93 (0.51–1.71)	0.44

^1^ Adjusted for age and diabetes. ^2^ Adjusted for age, sex, smoking history, body mass index, hypertension, and diabetes. RAE, retinol activity equivalents; CI, confidence interval. Nutrient intake was used with energy-adjusted values.

## Data Availability

The datasets used and analyzed during the current study are available from the corresponding author on reasonable request. The data are not publicly available due to personal data of the participants.

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
