# Peer review of "Association of Dietary Nutrient Intake with Early Age-Related Macular Degeneration in Japanese-Americans"

_metabolites, 2021, doi:10.3390/metabo11100673_

Round 1

Reviewer 1 Report

The authors describe an interesting finding on diet and AMD in a Japanese-American population. The major finding is that those migrating to the USA from Japan increase their animal and SFA intake, promoting the development of early AMD. In contrast, those using plan-based fats are less likely to develop early AMD. This is a useful addition to the body of work on nutrition and AMD.

The manuscript would benefit from some adjustments and clarifications.

1) line47: copper is there to avoid zinc-induced copper deficiency and not as a supplement for AMD. this could be clarified; otherwise, people might deduce that copper supplementation is a key event.

2) Paragraph starting at Line51 would benefit from clarification.

3) Table 1 lists confounding factors, from which only diabetes and age are significant. Still, the multivariate analysis takes all parameters into account. While this could be correct, it needs to be justified why all, not just the significant confounders are used so there is no confusion.

4) Line 78 should define shortly what early AMD. It is stated in the methods at the end of the draft only.

5) Line 107, why all confounders need to be clear.

6) Paragraph starting at line 160 should be clarified. Especially true for the last sentence as it reads as if the authors would be doubting their own results.

7) Line 233: is it possible that the reason the values for other components are not showing previously reported outcomes is that these are estimated and not measured values? If this is possible please state it.

8) Line 250 it should be "centred on the macula".

Reviewer 2 Report

the present paper presents an interesting analysis of the relation between nutrients intake and AMD development in a well defined selected group. The experimental protocol is clearly described and results are deeply discussed. The conclusions are convincing and well integrated with previous results.
